# MAtt: A Manifold Attention Network for EEG Decoding

**Yue-Ting Pan**     **Jing-Lun Chou**     **Chun-Shu Wei**
National Yang Ming Chiao Tung University, Hsunchu, Taiwan
`wei@nycu.edu.tw`

## Abstract

Recognition of electroencephalographic (EEG) signals highly affect the efficiency of non-invasive brain-computer interfaces (BCIs). While recent advances of deep-learning (DL)-based EEG decoders offer improved performances, the development of geometric learning (GL) has attracted much attention for offering exceptional robustness in decoding noisy EEG data. However, there is a lack of studies on the merged use of deep neural networks (DNNs) and geometric learning for EEG decoding. We herein propose a manifold attention network (MAtt), a novel geometric deep learning (GDL)-based model, featuring a manifold attention mechanism that characterizes spatiotemporal representations of EEG data fully on a Riemannian symmetric positive definite (SPD) manifold. The evaluation of the proposed MAtt on both time-synchronous and -asyncronous EEG datasets suggests its superiority over other leading DL methods for general EEG decoding. Furthermore, analysis of model interpretation reveals the capability of MAtt in capturing informative EEG features and handling the non-stationarity of brain dynamics. Source codes are available at `https://github.com/CECNL/MAtt`.

## 1 Introduction and related works

A brain-computer interface (BCI) is a type of human-machine interaction that bridges a pathway from brain to external devices. Electroencephalogram (EEG), a non-invasive neuromonitoring modality with high portability and affordability, has been widely used to explore practical applications of BCI in the real world [1, 2, 3]. For instance, disabled users can type messages through an EEG-based BCI that recognizes the steady-state visual evoked potential (SSVEP) induced by flickering visual targets presented on a screen [4, 5, 6]. Stroke patients who need restoration of motor function undergo motor-imagery (MI) BCI-controlled rehabilitation as an active training [7, 8]. Most EEG-based BCI systems are designed to detect/recognize reproducible time-asynchronous or time-synchronous EEG patterns of interest, depending on the schemes of BCI [9]. For example, the MI EEG pattern is an endogenous oscillatory perturbation sourced from the motor cortex without an explicit onset time [10]. On the other hand, a time-synchronous EEG pattern is time-locked to a specific event. For example, the pattern of SSVEP is synchronized to the change of brightness on a flickering visual target. The efficiency of BCI systems largely relies on the accuracy and robustness of the EEG decoder. However, due to the low signal-to-noise ratio (SNR) [11] and non-stationarity [12] of EEG, translating perplexing EEG signals into meaningful information has been a grand challenge in the field.

Recent advances in deep learning (DL) have contributed to the rapid development of DL-based EEG decoding techniques [13]. DL models are capable of extracting features automatically according to given training data. Convolutional neural network (CNN) is one type of the most common DL models and has achieved remarkable performance in tasks such as image recognition and object detection [14, 15, 16]. CNN models newly designed for EEG decoding use convolutional kernels that

analogously function as conventional spatial and temporal filters but with extra flexibility to optimize the transformation of EEG data automatically through model training [17, 18, 19]. In addition to the fast growth of DL-based EEG decoders, geometric learning (GL) approaches, mostly based on Riemannian geometry (RG), have been adopted in the field of BCI [20]. RG is a type of non-Euclidean geometry that has a different interpretation of Euclid's fifth postulate (i.e. parallel postulate) [21]. In GL, geodesic between points on the manifold is a critical feature for classification tasks in BCI. The power and spatial distribution of a segment of multi-channel EEG signals can be coded into a covariance matrix that is symmetric positive definite (SPD) in general. The use of Riemannian geometry allows mapping of EEG data directly onto a Riemannian manifold where Riemannian metrics are insensitive to outliers and noise [22, 20]. RG can also avoid swelling effect [23], which is a common issue when employing Euclidean metric. Furthermore, metrics on Riemannian manifold have several types of invariance properties [24, 22], which make the model have higher generalization capability to complex EEG signals. In 2010, Barachant et al. [25] proposed Minimum Distance to Mean (MDM) that maps target EEG data onto the SPD manifold to find the nearest class center. Later on, they developed TSLDA [26] that projects data from the manifold to a specific tangent space where Euclidean classifiers are applicable. RG-based classification for EEG decoding has shown extra robustness as the relationship between data samples can be stably preserved, leading to success in recent data competitions in the BCI field such as 'DecMEG2014'[1] and the 'BCI challenge'[2].

The nascent field of geometric deep learning (GDL) [27] has expanded by emerging techniques to generalize the use of deep neural networks to non-Euclidean structures, such as graphs and manifolds. Efforts have been made to transitioning useful operations from Euclidean to Riemannian spaces, including convolution [28, 29, 30], activation function [28, 29], batch normalization [31, 32], that facilitate the ongoing development of GDL tools. SPDNet [28] is a Riemannian network for non-linear SPD-based learning on Riemannian manifolds using bi-linear mapping that mimics Euclidean convolution for visual classification tasks. ManifoldNet [29] offers high performance in medical image classification with manifold autoencoder. [33] characterizes 3D movement via the manifold polar coordinate with a geodesic CNN. [27] performs convolution on the manifold as a generalization of local graph or manifold pseudo-coordinate for vertex classification on graph and shape correspondence task. In contrast of the vast develop of GDL in many other scientific fields, only few studies focus on decoding EEG data with a merge use of GL and DL. [34] proposed a network architecture that integrates fusion of Euclidean-based module and manifold-based module with multiple LSTM and attention structures to extract spatiotemporal information of EEG. [35] proposes a Riemannian-embedding-banks method that separates the entire embeddings into multiple sub-problems for learning spatial patterns of MI EEG signals based on the features extracted from the SPDNet. [36] combines federated learning and transfer learning on Riemannian manifold using the spatial information of EEG. [37] proposes deep optimal transport on the manifold to minimize the cost of domain adaptation from the source domain to the target domain. [38] extracts multi-view representations of EEG. These studies have established cornerstones toward the field of future GDL for EEG decoding, but the increment of performance is yet marginal. Most of the above-mentioned techniques can not map the temporal information of EEG onto the manifold, or still rely on Euclidean tools to handle EEG features. We herein propose a manifold attention network, a novel GDL framework, which maps EEG features on a Riemannian SPD manifold where the spatiotemporal EEG patterns are fully characterized. The main contributions of the present study are the following:

- a manifold attention network proposed for decoding general types of EEG data.
- a lightweight, interpretable, and efficient GDL framework that is capable of capturing spatiotemporal EEG features across Euclidean and Riemannian spaces.
- an empirical validation of our proposed model demonstrating its generalizable superiority over leading DL approaches in EEG decoding.
- neuroscientific insights interpreted by the model that not only echo prior knowledge but also offer a new look into the dynamical brain.

This article is organized as follows: we first brief the essential background of RG and manifold attention mechanism; next, we leverage the proposed MAtt architecture with details of model design; we then validate our proposed model experimentally; lastly, we interpret our proposed model with neuroscientific insights. Our source code is released at `https://github.com/CECNL/MAtt`.

---

[1]DecMEG2014: https://www.kaggle.com/competitions/decoding-the-human-brain/leaderboard

[2]BCI challenge: https://www.kaggle.com/c/inria-bci-challenge

## 2 Preliminary

A manifold is a generalization of curves and surfaces in Euclidean space. It is a topological space that can be locally regarded as an open set in Hilbert space. If a manifold is equipped with a differential structure (i.e. a collection of charts with transition maps defined on the overlaps of the charts), it is called a differential manifold [39]. Riemannian geometry is the study of differential manifolds equipped with a Riemannian metric. We focus on the symmetric positive definite (SPD) manifold, which allows us to directly manipulate manifold-valued data on the manifold. The spatial information of an EEG signal can be represented as a specific covariance matrix, which encodes the relationships between channels and is a critical representation for understanding EEG signals. However, the solution of the Riemannian mean does not have a closed form when the manifold is equipped with an affine invariant metric (AIM), so we need to calculate the approximate mean in an iterative manner [25, 29] until convergence conditions are satisfied. Riemannian mean can be computationally expensive in deep learning due to its high complexity. Therefore, we use an approximation based on the Log-Euclidean metric [24] as described below.

### 2.1 Notations

$GL(n, \mathbb{R}) := \{A \in \mathbb{R}^{n \times n} \mid determinant(A) \neq 0\}$ is a general linear group, which is the set of all real non-singular sqaure matrices. $(\mathcal{M}, g)$ denotes connected Riemannian manifold. $Sym(n) := \{S \in M_{n \times n}(\mathbb{R}) \mid S^T = S\}$ is the space of all $n \times n$ real symmetric matrices, where $M_{n \times n}(\mathbb{R})$ specifies the space of all real square matrices, $(.)^T$ is the *transpose* operator, and $Sym^+(n) := \{P \in M_{n \times n}(\mathbb{R}) \mid P = P^T, v^T P v > 0, \forall v \in \mathbb{R}^n - \{0\}\}$ is the set of all $n \times n$ symmetric positive definite(SPD) matrices. $< A, B >_F$ means the Frobenius inner product, defined as $Tr(A^T B)$, where $Tr(.)$ is the *trace* operator. $Log(.)$ and $Exp(.)$ are the *principle logrithm* operator for SPD matrix [40] and *exponential* operator for symmetric matrix respectively. Both of them can be computed using the *orthogonal diagonalization*. $Exp : Sym(n) \mapsto Sym^+(n)$, an operator maps a symmetric matrix $S \in Sym(n)$ to $Sym^+(n)$ by:

$$Exp(S) = V diag(exp(\sigma_1), ..., exp(\sigma_n))V^T$$

where $V$ is the matrix of eigenvectors of $S$.

The inverse projection of $Exp$ operation is $Log$ operator: $Log : Sym^+(n) \mapsto Sym(n)$ is an operator that maps a SPD matrix $P \in Sym^+(n)$ to $Sym(n)$ by:

$$Log(P) = U diag(log(\sigma_1), ..., log(\sigma_n))U^T \tag{1}$$

where $U$ is the matrix of eigenvectors of $P$, since $P \in Sym^+(n)$, $\sigma_i > 0, i = 1, ..., n$

### 2.2 Log-Euclidean metric

Log-Euclidean metric (LEM) offers an elegant, analogous, and efficient generalization to calculate the center on the SPD manifold than the affine-invariant metric (AIM) [24, 41]. LEM is a bi-invariant metric on the Lie group on the SPD manifold [24]. The geodesic distance from $P_1$ to $P_2$ on the $Sym^+(n)$ is also given by [24]:

$$\delta_L(P_1, P_2) = \|Log(P_1) - Log(P_2)\|_F \tag{2}$$

Furthermore, we can also define the Log-Euclidean mean($\mathcal{G}$) via the Log-Euclidean distance:

$$\mathcal{G}(P_1, ...P_k) = \underset{P \in Sym^+(n)}{\arg\min} \sum_{l=1}^{k} \delta_L^2(P, P_l)$$

where $P_1, ..., P_k \in Sym^+(n)$. Fortunately, the solution to the formula above has a closed form to follow, given by [42]:

$$\mathcal{G} = Exp\left(\frac{1}{k} \sum_{l=1}^{k} Log(P_l)\right)$$

We utilizes the weighted Log-Euclidean mean that is endowed with different weights in different $P_l$ in our work. We denote the weight of each $P_l$ as $w_l$, where $\forall l \in \{1, 2, ..., k\}$. Here, $\{w_l\}_{l=1}^{k}$

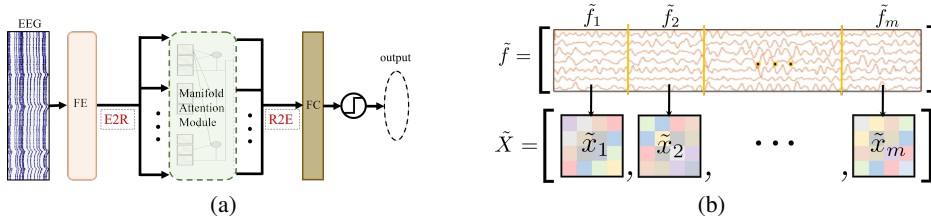

(a)                                    (b)

Figure 1: (a) The overview of the proposed model architecture. (b) E2R operation: split latent feature into several epochs, and convert each one to a specific SPD matrix.

satisfies the *convexity constraint* definition (i.e. $\sum_{l=1}^{k} w_l = 1$, and $w_l > 0$). The definition and the corresponding weighted Log-Euclidean mean can be defined and derived as:

$$\mathcal{G}(P_1, ... P_k) = \arg\min_{P \in Sym^+(n)} \sum_{l=1}^{k} w_l \delta_L^2(P, P_l)$$

and

$$\mathcal{G} = Exp\left(\sum_{l=1}^{k} w_l Log(P_l)\right)$$

respectively.

## 3 Methodology

As shown in Figure 1(a), the architecture of MAtt includes components of the feature extraction (FE), the manifold attention module, transitioning from Euclidean to Riemannian space (E2R), and transitioning from Riemannian to Euclidean space (R2E).

### 3.1 Feature extraction of EEG signals

We adopt two convolutional layers to extract information of raw EEG signals, where the first convolutional layer performs spatial filtering to the multi-channel EEG signals and the second convolutional layer extracts spatiotemporal features. Our parameter setting follows [19].

### 3.2 From Euclidean space to SPD manifold (E2R operation)

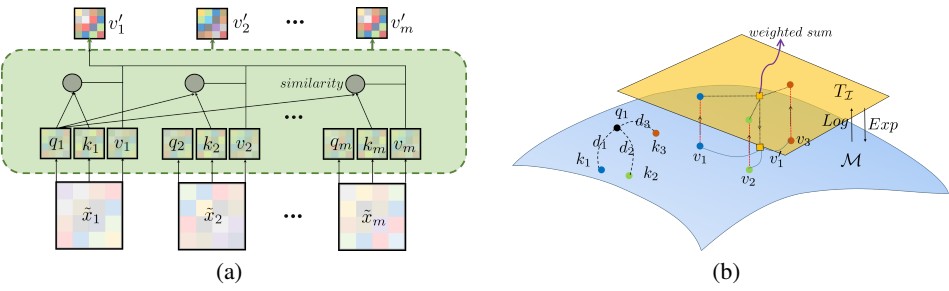

(a)                                    (b)

Figure 2: (a) The architecture of the proposed manifold attention module. $q_i, k_i, v_i$ refer to the query, key, and value of the $i^{th}$ input matrix $\tilde{x}_i$ respectively; $v_i'$ stands for the $i^{th}$ output of the proposed module. (b) Illustration of the operation of Log-Euclidean mean used in proposed module as $i = 1$ and number of epoch is 3; $q_i$ and $k_j$ refer to $i^{th}$ query and $j^{th}$ key respectively; $d_j$ denotes the distance between $q_1$ and $k_j$ on the SPD manifold $\mathcal{M}$; $T_{\mathcal{I}}$ refers to the tangent space based on identity matrix $\mathcal{I}$.

As illustrated in Figure1(b), we convert the embeddings from the feature extraction stage to the SPD data and map the feature embeddings from Euclidean space to the SPD manifold. Suppose $\tilde{f}$ denotes the embeddings after the feature extraction stage, we divide the whole embeddings into several epochs $\tilde{f}_1, \tilde{f}_2, ..., \tilde{f}_m$, and calculate the sample covariance matrix(SCM) of each $\tilde{f}_i, \forall i \in \{1, 2, ..., m\}$. By doing so, we get a sequence of covariance matrices that present the temporal information of the embeddings $\tilde{f}$ in the form of SPD data, called $SCM_{\tilde{f}_1}, SCM_{\tilde{f}_2}, ..., SCM_{\tilde{f}_m}$. After we get some datapoints, we do trace-normalization and add a small number $\epsilon$ on each main diagonal element for each $SCM_{\tilde{f}_i}$ (i.e. $\frac{SCM_{\tilde{f}_i}}{tr\left(SCM_{\tilde{f}_i}\right)} + \epsilon I$) where $i \in \{1, 2, ..., m\}$, $I$ is the identity matrix, and we set $\epsilon$ as 1e-5 in our source code. The resulting SPD sequence is denoted as $\tilde{X} = [\tilde{x}_1, \tilde{x}_2, ..., \tilde{x}_m]$. We add a small identity matrix to ensure that each $\tilde{x}_i$ is a well-defined SPD matrix.

### 3.3 Manifold attention module

**Forward procedure:** The input of this layer is a sequence of SPD data. The overview of the manifold attention module is illustrated in Figure 2(a). Motivated by [28] and [43], we capture the spatiotemporal information on the manifold. Suppose the module takes a sequence of SPD matrices $[\tilde{x}_1, \tilde{x}_2, ..., \tilde{x}_m]$, denoted as $\tilde{X}$. Herein we have the query, key, and value in the form of SPD matrices on the manifold [43]. We convert the $\tilde{x}_i$ to the $q_i, k_i$, and $v_i$ via bilinear mapping [28] and exploit non-linear and valid features from each segment. Suppose the shape of $\tilde{x}_i$ is $d_c \times d_c$, and $h_q, h_k$, and $h_v$ is the mapping from $\tilde{x}_i$ to $q_i, k_i$, and $v_i$ respectively. We have:

$$q_i = h_q(\tilde{x}_i; W_q) = W_q \tilde{x}_i W_q^T; \; k_i = h_k(\tilde{x}_i; W_k) = W_k \tilde{x}_i W_k^T; \; v_i = h_v(\tilde{x}_i; W_v) = W_v \tilde{x}_i W_v^T$$

where $\tilde{x}_i \in Sym^+(d_c)$, $W_q, W_k$, and $W_v \in \mathbb{R}^{d_u \times d_c}(d_u < d_c)$ denotes transformation matrices. Moreover, to make sure the output $q_i, k_i$, and $v_i$ are also SPD matrices, transition matrices $W_q, W_k$, and $W_v$ are constrained as row-full rank matrices.

After we got $q_i, k_i$, and $v_i$ by bilinear mapping, we define the similarity for measuring the $q_i$ and $k_j$ SPD matrices. In Euclidean space, there are several ways to define the similarity. A most common way is to use dot-product [43] to measure the similarity of query and key. However, our query, key, and value are SPD matrices instead of vectors as regular attention. We define the similarity based on the Log-Euclidean distance (Eq. 2) between query and key. Suppose we have $q_i$ and $k_j$, for some $i, j \in \{1, ..., m\}$. The similarity $sim(.)$ is a *strictly decreasing function* of distance $[0, \infty) \mapsto [0, 1]$ and is defined as: $sim(q_i, k_j) = \frac{1}{1+log(1+\delta_L(q_i, k_j))} := \alpha_{ij}$. Then, the attention matrix is:

$$\mathcal{A} = [\alpha_{ij}]_{m \times m}$$

We then use $Softmax$ function to shrink the range along the row direction, making values in row have *convexity constraint* property. The final attention probability matrix $\mathcal{A}'$ is:

$$\mathcal{A}' = Softmax(\mathcal{A}) = Softmax([\alpha_{ij}]_{m \times m}) = [\alpha'_{ij}]_{m \times m}$$

where $\alpha'_{ij} = \frac{exp(\alpha_{ij})}{\sum_{k=1}^{m} exp(\alpha_{ik})}, \forall i, j \in 1, \cdots, m$. Finally, we combine the attention probability matrix and $v_1, v_2, ..., v_m$ to get the final output $v'_1, v'_2, ..., v'_m$ and define the output $v'_i (\forall i = 1, 2, ..., m)$ via Log-Euclidean mean as:

$$v'_i = Exp\left(\sum_{l=1}^{m} \alpha'_{il} Log(v_l)\right)$$

The forward procedure of proposed manifold attention module is illustrated in Algorithm 1.

**Backward procedure:** In order to perform gradient descent parameter updating on the Riemannian manifold, we employed the Riemannian gradient descent method to update the parameters. The trainable parameters in this module are $W_q, W_k$, and $W_v$. We require the weight updated on Stiefel manifold [44, 28], denoted as $St(p, n) = \{X \in \mathbb{R}^{n \times p} | X^T X = \mathcal{I}_p\}$. Since our manifold attention module has a different mathematical architecture to those in Euclidean space, we herein extend Euclidean gradients onto a Riemannian space. To be precise, we expect our gradients on the Stiefel manifold to generate valid orthogonal weights. The Euclidean gradients of $W_q, W_k$, and $W_v$ in the attention module can be derived using the chain rule. Suppose the $\mathcal{L}$ is the loss, the query, key, and value generated in the manifold attention module are $q_i, k_i$, and $v_i \forall i = 1 \cdots m$ respectively, and the

---

**Algorithm 1** Manifold attention module

---

**Require: A sequence of SPD data $\{\tilde{x}_i\}_{i=1}^m$ , transformation matrices $W_q, W_k, W_v$**
 1: **for** $i = 1 : m$ **do**
 2: $\quad q_i = W_q \tilde{x}_i W_q^T; \ k_i = W_k \tilde{x}_i W_k^T; \ v_i = W_v \tilde{x}_i W_v^T$
 3: **end for**
 4: $\forall i, j \in \{1, \cdots, m\}, \mathcal{A} := [\alpha_{ij}]_{m \times m} = \frac{1}{1 + log(1 + \delta_L(q_i, k_j))}$
 5: $\mathcal{A}' = Softmax(\mathcal{A})$
 6: **for** $i = 1 : m$ **do**
 7: $\quad v_i' = Exp\left(\sum_{l=1}^m \alpha_{il}' Log(v_l)\right)$
 8: **end for**
 9: **return a sequence of SPD data $\{v_i'\}_{i=1}^m$**

---

output of the attention module is $v_i' \ \forall i = 1 \cdots m$ (All notations here are the same as in the Algorithm 1):

$$\nabla_{W_q}\mathcal{L} = \sum_{i=1}^m \frac{\partial \mathcal{L}}{\partial v_i'} \frac{\partial v_i'}{\partial W_q} = \sum_{i=1}^m \frac{\partial \mathcal{L}}{\partial v_i'} \frac{\partial v_i'}{\partial q_i} \frac{\partial q_i}{\partial W_q} = \sum_{i=1}^m \frac{\partial \mathcal{L}}{\partial v_i'} \left(\sum_{j=1}^m \frac{\partial v_i'}{\partial \alpha_{ij}'} \frac{\partial \alpha_{ij}'}{\partial q_i}\right) \frac{\partial q_i}{\partial W_q}$$

$$\nabla_{W_k}\mathcal{L} = \sum_{i=1}^m \frac{\partial \mathcal{L}}{\partial v_i'} \frac{\partial v_i'}{\partial W_k} = \sum_{i=1}^m \frac{\partial \mathcal{L}}{\partial v_i'} \left(\sum_{j=1}^m \frac{\partial v_i'}{\partial k_j} \frac{\partial k_j}{\partial W_k}\right) = \sum_{i=1}^m \frac{\partial \mathcal{L}}{\partial v_i'} \left(\sum_{j=1}^m \frac{\partial v_i'}{\partial \alpha_{ij}'} \frac{\partial \alpha_{ij}'}{\partial k_j}\right) \frac{\partial k_j}{\partial W_k}$$

$$\nabla_{W_v}\mathcal{L} = \sum_{i=1}^m \frac{\partial \mathcal{L}}{\partial v_i'} \frac{\partial v_i'}{\partial W_v} = \sum_{i=1}^m \frac{\partial \mathcal{L}}{\partial v_i'} \left(\sum_{j=1}^m \frac{\partial v_i'}{\partial v_j} \frac{\partial v_j}{\partial W_v}\right)$$

The way of the attention module to impose the Euclidean gradient to the Stiefel gradient on the manifold [44] is: Suppose $\nabla_{W_v}\mathcal{L}$ is a $d_u \times d_c$ matrix and $W_v$ is on the manifold. Then the projection of $\nabla_{W_v}\mathcal{L}$ onto the normal space, which is the orthogonal complement of the tangent space based on the tangent point $W_v$ is:

$$\pi_N\left(\nabla_{W_v}\mathcal{L}\right) = W_v\left(W_v^T \nabla_{W_v}\mathcal{L}\right)_{sym} = W_v\left(\frac{W_v^T(\nabla_{W_v}\mathcal{L}) + (\nabla_{W_v}\mathcal{L})^T W_v}{2}\right) \quad (3)$$

where the $D_{sym}$ is defined as $\frac{D + D^T}{2}$. The tangent component of the $St(d_c, d_u)$ can be defined as the subtraction between the Euclidean gradient and normal component $\pi_N\left(\nabla_{W_v}\mathcal{L}\right)$ from Eq. 3:

$$\tilde{\nabla}_{W_v}\mathcal{L} = \nabla_{W_v}\mathcal{L} - \pi_N\left(\nabla_{W_v}\mathcal{L}\right) = \nabla_{W_v}\mathcal{L} - W_v\left(\frac{W_v^T(\nabla_{W_v}\mathcal{L}) + (\nabla_{W_v}\mathcal{L})^T W_v}{2}\right)$$

Now we have the Stiefel gradient $\tilde{\nabla}_{W_v}\mathcal{L}$, which is the tangential direction that we update the parameters. We use the *retraction* operation, which is a smooth mapping to map the final weight back to the $St(d_c, d_u)$. Finally, We have the updated weight:

$$W_v^{(new)} = \Gamma(W_v - \lambda \tilde{\nabla}_{W_v}\mathcal{L})$$

where $\lambda$ is the learning rate, $\Gamma$ is the retraction operation defined in the QR decomposition [45]. The backward procedure for $W_q$ and $W_k$ are the same. For more concepts about the optimization in Riemannian geometry, we recommend readers refer to [44, 45].

As shown in Figure 2 (b), the output $v_1'$ of our attention module can be comprehended as a projection that translates the weighted sum, on the tangent space, of three different matrices encoded by three different epochs and corresponding attention scores $\alpha_{11}', \alpha_{12}', \alpha_{13}'$ (or weights) to a specific representative matrix $v_1'$ on SPD manifold. Herein the weights for the weighted sum on the tangent space is assigned by its query matrix $q_1$ and corresponding keys $k_1, k_2, k_3$ to generate the relevance score between $q_1$ and $k_1, k_2, k_3$ [43, 46].

## 3.4 From Riemannian manifold to Euclidean space (R2E) and loss layers

After passing through the attention module, the ReEig layer is used to imitate the ReLU function. However, unlike the ReLU function, which sets a threshold to the value of the input, the ReEig layer sets a threshold to the eigenvalue of the input, as defined in [28]. The R2E operation aims to map the SPD data back to Euclidean space for the final classification, which is composed of a logarithmic layer and regular flatten layer in [28]. The logarithmic layer is a common technique in geometric deep learning for projecting SPD data to Euclidean space, allowing us to reduce the manifold to a flat space using the $Log(.)$ operation. We apply the $Log(.)$ operation to the output from the attention module layer, $v'_1, v'_2, ..., v'_m \in Sym^+(d_u)$. Denote the R2E operation as $h_L \colon Sym^+(d_u) \mapsto \mathbb{R}^{d_u \times (d_u+1)/2}$:

$$ h_L(v'_i) = flatten\left(Log(v'_i)\right) = flatten\left(S(diag(log(\sigma_1), \cdots, log(\sigma_{d_u})))S^T\right) $$

where $S$ is the eigenvector matrix of $v'_i$, and $\sigma_1, \cdots, \sigma_{d_u}$ are the eigenvalues of $v'_i$. The $Log(.)$ operation is the same as Eq. 1, and the $flatten(A)$ operation flatten the upper triangle of the arbitrary symmetric matrix $A$.

Finally, we set a fully connected layer and regular softmax operation on embeddings after R2E operation. Suppose the output from the whole model stream is $\hat{y}$, the groundtruth is $y$, we define the loss $\mathcal{L}$ as the cross-entropy loss of $\hat{y}$ and $y$.

# 4 Experiments

Here we evaluate the proposed MAtt using both time-asynchronous and time-synchronous EEG data to give empirical evidence of the advantages. The performance in a general use for EEG decoding is compared against leading DL-based techniques. We incorporate the BCI Competition IV 2a Dataset (BCIC-IV-2a) [47] to assess the performance on time-asynchronous motor-imagery (MI) EEG decoding , the MAMEM EEG SSVEP Dataset II (MAMEM-SSVEP-II) [48] and the BCI challenge error-related negativity (ERN) dataset (BCI-ERN) [49] to assess the performance on time-synchronous SSVEP and ERN EEG decoding. Previous and current state-of-the-art DL-based models listed for comparison with MAtt include MBEEGSE [50], TCNet-Fusion [51], EEG-TCNet [52], FBCNet [53], SCCNet[19], EEGNet [17], and ShallowConvNet [18].

## 4.1 Datasets and preprocessing

**Data set I (MI)**: We employ the BCIC-IV-2a as a representative dataset providing time-asynchronous EEG data. The BCIC-IV-2a dataset is one of the most commonly used public EEG dataset released for the BCI Competition IV in 2008 [47]. It contains EEG recordings in a four-class motor-imagery task from nine subjects with two repeated session each on different days. During the task, the subjects were instructed to perform imagination of one of the four types of movements (right hand, left hand, feet, and tongue) for four seconds after an instructional cue. Each session consists of a total of 288 trials with 72 trials for each type of the motor imagery. The EEG signals were recorded by 22 Ag/AgCl EEG electrodes at the central and surrounding regions in a sampling rate of 250 Hz. We performed standard preprocessing procedures for the 22-channel EEG signals, including 1) Down-sampling from 256 Hz to 128 Hz, 2) Band-pass filtering at 4-38 Hz, and 3) Segmenting EEG signals at 0.5-4s (438 timepoints) of the onset of cue for each trial.

**Data set II (SSVEP)**: The MAMEM-SSVEP-II dataset was incoporated as the representative dataset of time-synchronous EEG data. It consists of EEG recordings from 11 subjects performing a SSVEP-based task where the subjects were instructed to gaze at one of the five visual stimuli flickering at different frequencies (6.66, 7.50, 8.57, 10.00, and 12.00 Hz) for five seconds. Each subject executed five runs of five cue-based trials for each of the five stimulation frequencies. The EEG signals were acquired using the EGI 300 Geodesic EEG System (GES 300) equipped with 256 channels in a sampling rate of 250 Hz. There are 5 sessions in this dataset. We assigned the early blocks (session 1, 2, and 3) as the training set, session 4 as validation set and the remaining (session 5) as the test set to maximize the efficacy of the cross-session validation. The preprocessing procedures for this dataset were 1) Band-pass filtering at 1-50 Hz, 2) Selecting eight channels (PO7, PO3, PO, PO4, PO8, O1, Oz, and O2) in the occipital area, the location of visual cortex, and 3) Segmenting each trial into four 1-second segments at 1s-5s of the onset of cue, yielding a total of 500 trials of 1-second 8-channel SSVEP signals for each subject, thus the time length of input EEG data is 125.

Table 1: Performance comparison between MAtt and baseline DL methods on MI (BCIC-IV-2a), SSVEP (MAMEM-SSVEP-II), and ERN (BCI-ERN) datasets. Bold fonts mark the highest overall performance (MI, SSVEP: accuracy, ERN: AUC). We adopted Wilcoxon-sign rank test with Bonferroni correction to perform the multiple comparison between all models. The statistical test result is available in Appendix.

| Models | MI | SSVEP | ERN |
|---|---|---|---|
| ShallowConvNet | 61.84±6.39 | 56.93±6.97 | 71.86±2.64 |
| EEGNet | 57.43±6.25 | 53.72±7.23 | 74.28±2.47 |
| SCCNet | 71.95±5.05 | 62.11±7.70 | 70.93±2.31 |
| EEG-TCNet | 67.09±4.66 | 55.45±7.66 | **77.05±2.46** |
| TCNet-Fusion | 56.52±3.07 | 45.00±6.45 | 70.46±2.94 |
| FBCNet | 71.45±4.45 | 53.09±5.67 | 60.47±3.06 |
| MBEEGSE | 64.58±6.07 | 56.45±7.27 | 75.46±2.34 |
| MAtt | **74.71±5.01** | **65.50±8.20** | 76.01±2.28 |

**Data set III (ERN)**: Another time-synchronous BCI Challenge ERN dataset (BCI-ERN) is adopted to validate the capacity of presented MAtt. The BCI-ERN dataset [49] was employed in the BCI Challenge on Kaggle [3]. Twenty-six subjects participated in a P300-based BCI spelling task and measured the ERN, a type of event-related potential (ERP), elicited from the feedback of erroneous input displayed by the BCI speller. The aim of the experiment was to detect and determine the type of signal perturbation evoked by the erroneous feedback from a BCI speller to improve and assess the robustness. Therefore this task is a binary-and-imbalanced decoding task because the BCI speller detects more amount of correct inputs than the amount of erroneous inputs. EEG recordings were recorded by 56 Ag/AgCl EEG electrodes (VSM-CTF compatible system) whose region followed the extended 10-20 system in sampling rate of 600 Hz. There are five sessions (60 trials for the first four sessions and 100 trials for the last session) per subject, and the duration of a single EEG trial is 1.25 seconds. We used the 16 subjects released in the early stage of the Kaggle competition. The preprocessing steps include 1) Downsampling from 600 Hz to 128 Hz and 2) Band-pass filtering at 1-40 Hz. Each trial has a size of 56 channels by 160 timepoints after the preprocessing step.

**Model training and validation:** A series of experiments were conducted to evaluate the performances of the MAtt against other EEG decoders with the context of real-world BCI usage taken into account. In the real-world usage of BCI, a user usually needs to go through a training session for collecting a sufficient amount of individual EEG data for training the decoding model before executing the BCI system. To stick with the practical scenario, we performed an individual training scheme where a chunk of trials within a subject are assigned to the training set and the left-over trials within the same subject are used for testing [19, 53]. For the BCIC-IV-2a dataset, we used the first session of a subject to the training set where one out of eight was used for validation for MAtt with $m = 3$. The model with the lowest validation loss within 350 iterations was used for testing on the second session of the same subject. For the MAMEM-SSVEP-II/BCI-ERN dataset, we assigned the first four sessions of a subject to the training set where one out of four was used for validation for MAtt with $m = 7/m = 3$. The model with the lowest validation loss within 180/130 iterations was used for testing on the fifth session of the same subject. The classification performances for BCIC-IV-2a and MAMEM-SSVEP-II datasets were estimated by the mean accuracy across ten repeats for each subject. On the other hand, we use the same criterion as [17], herein the AUC score [54] is adopted to estimate the performance of BCI-ERN dataset due to the imbalanced issue.

### 4.2 Performance comparison

We validate the performance of MAtt against other leading methods. The criteria of selecting the baseline model collection here is based on: 1) code availability and completeness and 2) solid evaluation (e.g. cross-session) without additional auxiliary procedures (e.g. manual feature extraction, data augmentation, pre-trained model, etc). As shown in Table 1, MAtt outperforms all other leading DL methods on both time-synchronous (SSVEP) and -asynchronous (MI) EEG decoding. However, for the ERN dataset, MAtt seconds to the EEG-TCNet with 1% deficit but is at least 7% higher than EEG-TCNet on other two kinds of EEG decoding tasks. As most DL-based EEG decoders are

---

[3]BCI challenge: https://www.kaggle.com/c/inria-bci-challenge

Table 2: Overall accuracy (%) on BCIC-IV-2a and MAMEM-SSVEP-II, and overall AUC score on BCI-ERN (%) with parts within the model appended. FE: feature extractor; MA: manifold attention module; SA: self-attention module.

| Parts appended | MI | SSVEP | ERN |
|---|---|---|---|
| FE | 26.08±0.70 | 20.18±1.11 | 73.40±2.27 |
| MA | 60.73±5.80 | 30.51±2.57 | 59.47±3.56 |
| FE+SA | 49.19±2.72 | 22.91±2.00 | 63.77±1.71 |
| FE+MA (proposed) | **74.71**±5.01 | **65.50**±8.20 | **76.01**±2.28 |

designed for a single type of EEG decoding task due to high variability between different types of EEG data [53, 52, 19], the experimental results suggest that MAtt provides high-performancee EEG decoding across types of EEG data. Our study attests the robustness of proposed MAtt, which has strong generalization capacity to adapt general types of EEG data compared to other leading DL models.

### 4.3 Ablation study

We assess the significance of each of the major component in MAtt via a series of ablation analysis. As shown in Table 2, the accuracy reduces if we only use one component in our MAtt to do the classification. However, the combination of our proposed manifold attention module and feature extractor (FE+MA) achieves the best accuracy on all datasets. This implies that there is no redundant component in our proposed MAtt, and each part is needed for its non-negligible functions. The feature extractor aims to denoise and preprocess the EEG signals, and the attention module focus on integrating the preprocessed EEG signals and capturing underlying dynamics in the latent features. We can further compare the result between FE+SA and FE+MA to check the necessity of MA: the performance of FE+MA significantly outperform the regular self-attention based FE+SA on all validation EEG datasets.

### 4.4 Model interpretation

Through analysis for the interpretation of the proposed model, MAtt, we are able to uncover the underlying characteristics learnt from the data. Figure 3 illustrates the gradient response for MI EEG decoding across channel and across time. We can see left/right hand MI responses are strong at C4 and C3 corresponding to right/left motor cortices that control the lateral motor functions of the contralateral side of the body [57]. Both feet and tongue MI, that are not lateral movements, presents strong responses at CPz above the midline of motor cortex. The spatial distribution clearly exhibit asymmetric pattern for left/right hand MI and symmetric pattern for feet/tongue MI. The temporal information is available in Figure 4, where all four types of MI induce strong response at mu band

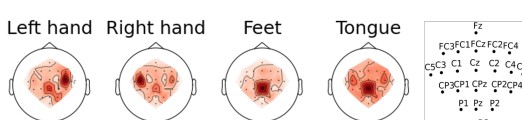

Figure 3: Spatial topomaps for the mean absolute gradient response (computed as in [55, 56]) across time from the visualization of the model S3 in the BCIC-IV-2a dataset for the four motor-imagery classes (left hand, right hand, feet, and tongue). Dark red marks the brains region presenting strong gradient activation at C4 (over right motor cortex) for the left hand, C3 (over left motor cortex) for the right hand, CPz (over motor cortex) for the feet and the tongue motor imagery.

around 10 Hz. This result is in line with the well-known association between motor function and mu rhythm in EEG recordings [57]. The visualization of our model for SSVEP decoding is exhibit in the appendix.

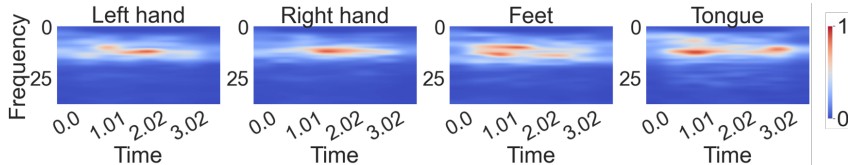

Figure 4: Time-frequency spectrograms from the gradient-based visualization (computed as in [55, 56]) of the model S3 in the BCIC-IV-2a dataset for the four motor-imagery classes (left hand, right hand, feet, and tongue). Strong response of motor imagery is marked by dark red at specific frequency bands and time intervals. Increased response of motor imagery is found at mu band ( 10 Hz) for all classes. The strong response of left/right hand motor imagery occurs at 1-2 seconds, the feet motor imagery is most vivid at 0.5-1 seconds, and the tongue motor imagery induced two peaks at 1 second and 3 seconds.

Figure 5 depicts the distribution of attention scores across epochs between the (a) MI, (b) SSVEP, and (c) ERN EEG signals. Here, the attention score refers to the average of the relevance score of an attention network, as described in [43], and has been applied to interpreting an attention-based EEG decoder [46]. For MI EEG signals, attention score is the highest at the first epoch and decreases in the following epochs, which implies that the beginning of the motor imagery

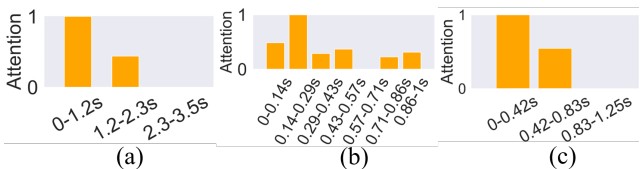

Figure 5: The distribution of attention scores across epochs within a trial based on the model interpretation for a single subject from the three datasets: (a) BCIC-IV-2a; (b) MAMEM-SSVEP-II; and (c) BCI-ERN.

may contribute a higher importance determined by the manifold attention module. The profile of attention score for SSVEP EEG signals presents a similar traits that the earlier epochs relate to higher importance. For ERN EEG signals, attention score is the highest on the first two epochs. As we observe a consistency cross EEG datasets that higher attention scores lie in earlier epochs, this may infers that the attention module relies largely on the similarity to the early stage of a trial, which is analogous to baseline correction, a major common procedure in conventional EEG signal processing [58]. This analysis reveals the capability of MAtt in handling the non-stationarity of the dynamical brain activity.

## Conclusion

We propose a manifold attention network as a novel GDL framework for both time-synchronous and -asynchronous EEG decoding. Using back propagation based on the Stiefel manifold, the proposed MAtt is capable of mapping EEG features onto a Riemannian manifold, where spatiotemporal EEG patterns are captured and characterized, within a lightweight architecture. The experimental results suggest the superiority of MAtt over current leading DL methods for both time-synchronous and -asynchronous EEG decoding. Spatial and temporal EEG patterns interpreted from the model are in line with prior neuroscientific knowledge and shed light on potential possibility of tracking the brain dynamics. In sum, our privileged method, MAtt, improves the SOTA performance of EEG decoding, and is expected to impact on GDL-based EEG processing with generalizable efficiency and robustness for future development of various BCI systems.

## 5 Acknowledgement

This work was supported in part by the National Science and Technology Council (109-2222-E-009-006-MY3, 110-2221-E-A49-130-MY2, and 110-2314-B-037-061) and in part by the Higher Education Sprout Project of the National Yang Ming Chiao Tung University and Ministry of Education of Taiwan.

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
