# OpenReview forum: "MAtt: A Manifold Attention Network for EEG Decoding"
_NeurIPS.cc/2022/Conference — NeurIPS 2022 Accept_

### Official Review · Reviewer_5euM · 2022-06-28

**Rating:** 5
**Confidence:** 4
**Soundness:** 3 good
**Presentation:** 2 fair
**Contribution:** 2 fair

**Summary:**

The paper proposes a deep neural network architecture equipped with a Riemannian manifold attention mechanism, tailored for EEG decoding tasks. Specifically, following convolutional feature extractor layers, learned Euclidean space feature sequences are mapped to an SPD manifold where the attention module operates. Experiments are evaluated on the BCI Competition IV-2a and an SSVEP dataset. Performed model interpretation analyses reveal neurophysiology-consistent predictions performed by the proposed architecture.

**Questions:**

- The paper needs to expand their analyses both in terms of methods to compare against, as well as the datasets considered. Further supporting experiments on larger MI datasets are necessary (e.g., "EEG datasets for motor imagery brain-computer interface". Gigascience. 2017) to show consistency with the BCIC-IV-2a results.

- While the MI-BCI experiments are designed well with cross-session test set separation, the SSVEP dataset uses a held-out experiment block of the same session as the test set, which is not a strong evaluation of generalization performance when it comes to EEG. Due to this limitation, I suggest another supporting time-synchronous EEG decoding experiment to be considered. For instance one could also consider an alternative experimental setup with P300 event-related potential detection experiments?

- Please clarify what is done in the process of "analysis of model interpretation"? It is undefined throughout which explanation tools or methods from papers are used for this. Also Figure 3 does not really stand for a strong outcome of the experimental analyses. If one would look into such topographies for the other models, similar neurophysiologically-consistent results can be also observed. It is not clear what is the direct comparison here.

- Discussion of results are very short. Method comparisons are only performed with respect to plain convolutional EEG decoding architectures (also, comparisons to DeepConvNet from [36] are missing?). There are no ablations of parts of the architecture to truly demonstrate its significance. At the most basic level, how relevant is the Riemannian manifold transformation? How does a CNN equipped with an attention module perform without SPD manifold projection? There are existing models exploiting temporal attention mechanisms that one can compare against:
"An end-to-end CNN with attentional mechanism applied to raw EEG in a BCI classification task." Journal of Neural Engineering, 2021.
"An attention-based deep learning approach for sleep stage classification with single-channel eeg." IEEE Transactions on Neural Systems and Rehabilitation Engineering, 2021.

- Figure 4 can be simplified by removing half of the Frequency range (y-axis values) from the figures, since the data was already preprocessed with a 4-38Hz bandpass filter?

- Input dimensionality of EEG inputs to the network for both experiments should also be clearly stated, it was not clear.



**Limitations:**

Some methodological limitations are addressed in Sec 4.3.

**Strengths And Weaknesses:**

Strengths: The paper nicely combines the traditional Riemannian manifold EEG feature extraction methods with a deep attention mechanism. Methodology is explained and illustrated clearly.

Weaknesses: Main limitation of the paper is its shallow experimental evaluations. This makes the proposal of the paper very weak as a general EEG decoding framework.

---

> ### Author Response · Authors · 2022-08-02
> **Response to Reviewer 5euM**
>
> Thank you for your valuable comments that helped us improve this study. We have provided responses to all of the comments and questions as following. For the detailed responses, please refer to the response letter in the supplementary material.
> 1) Shallow experiment: We have addressed the reviewers’ comments and made major revisions including additional experiments with additional baseline models and benchmark datasets. Kindly refer to Section 4.1 and Table 1 in the paper.
> 2) Cross-session validation: We agree with the reviewer that cross-session validation can better evaluate the robustness of performance across time. Although, in our study, only the BCIC-IV-2a dataset has cross-day sessions, we managed to applied cross-session validation for other datasets as we assigned the early blocks as the training set and the remaining as the test set to maximize the efficacy of the cross-session validation. We have added the description addressing this point in Section 4.1, second paragraph. Meanwhile, we appreciate the suggestion of incorporating another time-synchronous EEG dataset and thus an additional benchmark dataset has been added into the result as in Section 4.1.
> 3) Model interpretation: We appreciate the questions regarding the model interpretation that help us realized that additional explanation is needed. The description of the analysis of model interpretation is inserted in the caption of Figure 3, 4 and in Section 4.3.
> 4) Discussion and ablation analysis: Thank you for pointing out the weakness of the evaluation. We have added additional models and dataset for comparing the performance of our mAtt as in the response to the previous comment.
> 5) Figure 4: Thanks for the reminder. We addressed the reviewers' suggestion and revised the Figure 4 in the manuscript.
> 6) Input dimensionality: Thanks for the suggestion. Please kindly refer to appendix A.9 to check the EEG input size after the preprocessing.

---

> > ### Comment · Reviewer_5euM · 2022-08-03
> > **Thanks to the authors for their responses**
> >
> > I've read the authors' rebuttal, and I'd like to thank the authors for their efforts.
> >
> > Additional comparisons to other baseline models and benchmark datasets (Q1) add quite some value in my opinion. Particularly it shows the model's capability to extend to different EEG decoding tasks.
> >
> > I still believe that the interpretation analyses are a bit vague. Indeed, the authors clarified their descriptions for this section, and one can now clearly understand what is the approach used here. It is clear that these results are in line with neurophysiological evidence. However my point is also to explain (and show visually) what is different in Figures 3 & 4 with the proposed method, than others? Could we achieve similar visualizations with existing methods that mAtt compares against (I'd guess so)? E.g., are there particular subjects that these visual results become more noise-robust for mAtt?
> >
> > Overall, I'm increasing my final score in the light of the additional (supportive) experimental results.

---

> > > ### Author Response · Authors · 2022-08-09
> > > **Thanks to Reviewer 5euM for the responses**
> > >
> > > We appreciate the reviewer's comment and suggestion for us to compare the neurophysical visualization across models. We would like to clarify that the visualization analysis conducted in our study was to verify if the model captures neurophysiological patterns from the EEG data. Therefore, we actually see meaningful patterns learnt by all of the models that provides satisfactory classification accuracy. Please find the additional comparison we have conducted regarding this question in the next paragraph. Similar patterns are found by different models.
> > > We take BCI2a dataset and use subject3 to compare the visualzation of mAtt with the model (SCCNet) that has the highest acc among all baseline models. The table (see 'NIPS2022_response_letter_discuss.pdf' in the supplementary materials) depicts the topolplot and time-frequecny spectrogram of sub3 of SCCNet and mAtt.
> > >
> > > As shown in figure (see 'NIPS2022_response_letter_discuss.pdf' in the supplementary materials), there are vivid neurophysiological differences between the topoplots of SCCNet and mAtt. For example, when the subject was asked to do the right-hand imagination, the response of the left motor cortices would be strong in both models. However, there are some non-standard patterns shown on the right motor cortices and the middle line on the scalp on SCCNet. The vivid and extensive patterns are distributed from the right motor cortices to the occipital region. By contrast, our proposed mAtt is relatively robust, only 2 channels (CPz, C4) generate slight non-standard patterns. The analogous circumstances are shown in different classes. These visualization figures may exhibit the robustness of the proposed mAtt against the non-standard patterns. When it comes to the time-frequency spectrogram, for two spectrograms on both models, the obvious activation on the mu band showed up for both models. but there exists some interesting phenomena on spectrograms for mAtt and SCCNet. We take the class 'Tongue' as an example, the activated but not vivid pattern showed up on about 3 seconds of the onset in the mu band (8-13 Hz) for SCCNet. In contrary, in our proposed mAtt, the pattern at the same location would be activated. This may imply our mAtt took this pattern into account as predicting the tongue label, and our experimental result in table 1 also depicts the superiority of mAtt. This suggests that our mAtt may capture important but ambiguous underlying EEG patterns.
> > >
> > > However, it is hard to justify if the visualization infers the robustness of the models because those patterns all support the predictivity of the corresponding model, based on the evidence so far. The accuracy of the model should be regarded as a confidence level for the neurophysiological pattern observed, which might be useful when it comes to exploring brain dynamics based on an explainable neural network model.

---

### Official Review · Reviewer_SYJH · 2022-07-12

**Rating:** 8
**Confidence:** 3
**Soundness:** 4 excellent
**Presentation:** 4 excellent
**Contribution:** 4 excellent

**Summary:**

The authors submitted a well-written and inspiring paper proposing various attention networks for decoding general types of EEG data. It was a pleasure to read it, and the author presented a solid validation of their submission with two open-source BCI datasets.

**Questions:**

Did the authors consider other BCI paradigms (ERP/P300, cVEP, etc.), which are not narrow-frequency bands localized as MI or SSVEP?

**Limitations:**

Within-subject training and testing is a classical and limited ML application to BCI. Did the authors consider testing their method in transfer learning or leave-one-subject-out cross-validation setting, in which most BCIs fail these days?

**Strengths And Weaknesses:**

The proposed approach is novel, inspiring, clearly explained, supported with supplementary code, and ideally validated. The strength and solid contribution to ML in BCI is a lightweight, interpretable, and robust GDL framework capturing spatiotemporal EEG features across Euclidean and Riemannian spaces.

---

> ### Author Response · Authors · 2022-08-02
> **Response to Reviewer SYJH**
>
> Thanks for comments to help us make our study better. We address the comment and suggestions and revised the manuscripts carefully.
>
> 1) Other BCI paradigms: We agree with the reviewer's opinion. We have added an ERN dataset and additional baseline models in our revised paper. Please kindly refer to the Table 1 in the paper.
> 2) Leave-one-subject-out validation: Thanks for the insightful comment. Although our study focuses on developing a novel GDL-based model capable of enhancing EEG decoding in multiple types of EEG data, we will definitely include LOSO validation in our future work to verify the performance of our proposed model in this setting. It would be extremely interesting to test the robustness of our Riemannian manifold framework on subject-to-subject transferring.

---

### Official Review · Reviewer_hJCd · 2022-07-15

**Rating:** 6
**Confidence:** 4
**Soundness:** 1 poor
**Presentation:** 1 poor
**Contribution:** 2 fair

**Summary:**

The paper proposes a manifold attention network (mAtt), that leverages the properties of the SPD manifold in order to perform spatiotemporal analysis on EEG signals. In order to achieve that, the authors introduce a specific covariance matrix representation of the spatial information of the EEG signal. Concretely, they calculate the sample covariance matrix for each of the embeddings derived from the output of two convolutional layers fed with EEG time series. Then, the sequence of the covariance matrices (which are SPD matrices) is fed into a manifold attention layer that applies the attention mechanism on the SPD manifold.

**Questions:**

The capturing of temporal information is not very clear. How creating covariance matrices of consecutive EEG time-frames is better for capturing temporal features than using simple LSTM or GRU layers that have demonstrated excellent performance on similar tasks? (Zhang, Dalin, et al. "Cascade and parallel convolutional recurrent neural networks on EEG-based intention recognition for brain-computer interface.")

SPD matrices have been widely used in medical data analysis. However, to our knowledge, the criteria for choosing a particular metric on the SPD manifold for a given application are vague. Therefore, it would be nice to see the reasoning behind the choice of the Log-Euclidean metric instead of other alternatives (i.e. affine invariant metric Thanwerdas, Yann, and Xavier Pennec. "Is affine-invariance well defined on SPD matrices? A principled continuum of metrics."). Also, a comparative study demonstrating the advantages of one metric over the others would be an interesting addition to the paper.

The framework is tested on 9 subjects (asynchronous data) and 11 subjects (synchronous data). However, in order to be able to safely consider its generalization capabilities, it would be nice to evaluate how the framework performs given a bigger multi-subject dataset (i.e. EEG Motor Movement/Imagery Dataset from PhysioNet).

Fig 5 reveals some very interesting insights regarding neural dynamics with respect to the time duration. It would be interesting to see whether they can be verified by current neuroscientific knowledge.


**Limitations:**

Some of the limitations have been addressed in section 4.3.  Potential limitations that have not been addressed by the authors have been extensively discussed in the Questions section of this review.

**Strengths And Weaknesses:**

**Strengths**

The interpretability of the proposed framework is validated by the current neuroscientific knowledge. Concretely, Fig. 3 very nicely demonstrates brain regions corresponding to certain motor imageries of specific body parts.

Significance:
The proposed framework demonstrates non-trivial advances in terms of performance compared to sota methods.

**Weakness**

Clarity:
Line 114 there seems to be a plus (+) sign missing from the Sym(n) space.
Line 119 A clarification regarding what AIM stands for, followed by the corresponding literature reference, would be useful. (I assume it stands for affine invariant metric)
Fig. 2 More information is required in the caption.
Section 3.3  a very brief explanation of the terms query, key and value would give a nice ‘flow’ to the reading process of the paper, even though these terms have been extensively discussed in the referenced literature.

Originality:
The idea of representing EEG signals as covariance matrices and then projecting them on the SPD manifold has been widely used. The novelty of the approach lies mainly in the application of the attention mechanism on the SPD manifold. Therefore it would be interesting to see how the model performs when compared with very similar frameworks that exploit attention in different manners, i.e. Zhang, Guangyi, and Ali Etemad. "RFNet: Riemannian fusion network for EEG-based brain-computer interfaces."

---

> ### Author Response · Authors · 2022-08-02
> **Response to Reviewer hJCd**
>
> We appreciate your comments and suggestions that truly enhanced the quality of our paper. We have made a major revision accordingly, including appending notations of Fig 2, correcting the error in line 114, and adding a brief explanation of query, key, and value in Section 3.3. For the detailed responses, please refer to the response letter in the supplementary material. Brief responses are presented as following:
> 1) Capturing of temporal information: The meaning of capturing temporal information in our manifold attention module is similar to the regular attention module in [1]. We revised the description in the last paragraph of Section 3.3. The difference is we manipulate the sequence of EEG data on the manifold instead of regular Euclidean space. We first encode the feature segment into a specific covariance matrix, then we utilize the relevance calculation on two segments with each other to imply how relevance of these two segments on the manifold. Each attention score denotes a relevance score between two segments, and thus it can capture the temporal relation between time segments on the manifold. Moreover, the relevance score also used to calculate the final output (the output of our manifold attention) that is composed of a bunch of attention score and the corresponding time epoch information, can be considered as a combination of all epoch information. Therefore this module explores the temporal information on the manifold.
> We also provide other attention-based model to compare the result with mAtt in Table 1. In Table 1, we chose numerous temporal feature extraction methods such as attention, causal convolution, and nascent pooling to make a comparison (please kindly refer to Section 4.1). Moreover, the ablation study is listed in Section 4.2. Furthermore, the ablation study demonstrates the efficiency of our manifold attention module against the regular attention module.
> [1] Attention is all you need by Ashish Vaswani et al.
> [2] Cascade and parallel convolutional recurrent neural networks on EEG-based intention recognition for brain-computer interface by Dalin Zhang et al.
> [3] Deep learning-based electroencephalography
> analysis: a systematic review by Yannick Roy et al.
>
> 2) Choice of metric on the SPD manifold: We address the question and we also agreed that the criterion to choose a suitable metric on the SPD manifold is vague. the reason why we decide to choose this log-Euclidean metric can refer to appendix A.3: "...Moreover, the Riemannian metric has several properties..." part. Although the affine-invariant metric has elegant properties on the SPD manifold, such as congruence-invariant (kindly refer to appendix A.3). However, the price paid for its success is the high computational complexity when calculating the final output in our manifold attention module (Riemannian mean should be approximated in a recursive way, kindly refer to reference 7 and 8 in appendix). On the other hand, the calculation of mean based on the log-Euclidean metric has drastic reduction in computation time, but preserves the excellent theoretical properties (for more detailed introduction, kindly refer to 'Invariance Properties of Log-Euclidean Metrics' in [1]). We will list the comparsion between different metrics on SPD manifold in the future work. The comparison sounds interesting. Thanks the reviewer providing insightful perspectives.
> [1] Log-Euclidean Metrics for Fast and Simple Calculus Diffusion Tensors by Vincent Arsigny et al.
>
> 3) Bigger dataset: Thanks for the reviewer's constructive suggestion. We will expand the validation on different types of datasets in the future, such as the EEG Motor Movement/Imagery Dataset from PhysioNet. Please kindly refer to Section A.7 future work in appendix:
>
> 4) Neuroscientific insight: Thank you for noticing the neuroscientific insight obtained in our analysis. In Section 4.3, as we observe a consistency cross EEG datasets that higher attention scores lie in earlier epochs, this may infers that the attention module relies largely on the similarity to the early stage of a trial, which is analogous to baseline correction, a major common procedure in conventional EEG signal processing. Future work regarding this direction should involve novel design of experiment for collecting specific data and development of relevant analysis that can bridge the mechanism of the GDL-based model and the neuroscience to be explained. Please see our Future work in Appendix A.7.

---

> > ### Comment · Reviewer_hJCd · 2022-08-07
> > **Appreciate the response, increasing score**
> >
> > Thanks for addressing most of my questions. I’m happy with the responses and confident that the promised changes, particularly the Future work in Appendix A.8, will greatly improve this paper. I have changed my score to reflect this.

---

> > > ### Author Response · Authors · 2022-08-09
> > > **Thanks to Reviewer hJcD for the responses.**
> > >
> > > We greatly appreciate the reviewer's positive response to our revision and are delighted to see the score changed.

---

### Official Review · Reviewer_AD4a · 2022-07-17

**Rating:** 4
**Confidence:** 3
**Soundness:** 2 fair
**Presentation:** 2 fair
**Contribution:** 2 fair

**Summary:**

The proposed MAtt is a manifold attention-based model to capture EEG representations. I am not an expert in manifold and the 'geometric' in this paper is not have the same meaning in my domain (where geometric learning denotes graph neural networks). Therefore, please decrease the weight of my comments and consider more from other reviewers.

I will provide the comments as a general researcher with expertise in EEG.

**Questions:**

See Weaknesses above.

BTW, It's better to move Sec. 3.5 to Experiments (Sec. 4).

**Limitations:**

The authors discussed their limitations in Sec. 4.3.

**Strengths And Weaknesses:**

Strengths:

1. Considered both MI EEG and SSVEP. MI is a passive EEG while SSVEP is an event-evoked EEG. They are two key categories in EEG family.

Weaknesses:

1. The experimental results in Table 2 are kind of weird to me (lower than expected), so I checked the performances in STOA. The performance of MI-BCI competition IV-2a reported in the literature is higher than that reported in this work.

For example, [1] provides the accuracy of this dataset ranges from 76% to 79% (higher than the highest value in Table 2). [2] shows the accuracies ranges from 75% to 81%. Please check, compare, and explain.

[1] S-EEGNet: Electroencephalogram Signal Classification Based on a Separable Convolution Neural Network With Bilinear Interpolation, 2020.
[2] Advanced TSGL-EEGNet for Motor Imagery EEG-Based Brain-Computer Interfaces Lack of STOA baselines, 2021.

2. Lack of SOTA baselines. This manuscript only compared with four baselines which are published in 2017, 2018, 2019, and 2021 (preprint) separately. More recent and powerful baselines are expected to be included in Table 2.

3. In the Introduction, it is not clear why geometric deep learning is necessary in EEG classification. Manifold-based methods have boost performance in images is not a good idea for using the same methods in EEG.

---

> ### Author Response · Authors · 2022-08-02
> **Response to Reviewer AD4a**
>
> We are grateful for the comments and suggestions from the editors and the reviewers. We have revised the manuscript and have addressed issues in every comment.
> 1) Higher performance in other papers: The diverse performances of using the same EEG dataset across studies may source from the inconsistency of 1) validation schemes and 2) pre-processing procedures. The cross-session validation is a more rigorous evaluation for an EEG decoder without having temporally-adjacent EEG segments in training and testing set. The detailed response is available in the response letter in the supplementary material.
> 2) Lack of SOTA baselines: Thanks for pointing out the limitation of our work. We have addressed the reviewers’ comments and made major revisions and experiments such as comparison with additional methods accordingly in the manuscript. Kindly refer to the Table 1 to check in the paper.
> 3) GDL for EEG classification:
> Theoretically speaking, characteristics of EEG data suffer from several types of issues. Raw EEG signals are obtained from many reference electrodes on the scalp. The source of EEG signals will be mixed by many unexpected noises. Thus, raw EEG signals are sensor-space EEG signals. We have many types of methods to cope with these noise-source mixed circumstance. For example, independent component analysis (ICA). Appendix A.3 also demonstrates the strength of manifold-based metrics. The property (congruence invariance) of the Riemannian metric can deal with this issue to a certain degree [1]. Second, surveys on the geometric deep learning is rare applied in EEG decoding (refer to Section related work). Our study aims to exploit more interesting results (such as if the interpretability matches the knowledge in neuroscience) when it comes to geometric deep learning applied in BCI community. In our humble opinion, [2] would be a brilliant work for explaining the difference between the sensor-space and source-space MEG/EEG in the manner of mathematics, and demonstrates the strength of different kinds of no-Euclidean metrics on manifolds.
> Practically speaking, the ablation study in the Section 4.2 also demonstrates the superiority of our proposed manifold-based method. The presented manifold attention module would be effective in EEG decoding than Euclidean-based methods; second, the Table I in the paper exhibits the strength of our lightweight manifold-based mAtt.
> [1] Riemannian geometry for eeg-based brain-computer interfaces; a primer and a review. Brain-Computer Interfaces, 2017.
> [2] Manifold-regression to predict from MEG/EEG brain signals without source modeling, 2019.

---

> > ### Comment · Reviewer_AD4a · 2022-08-07
> > **Appreciate the response**
> >
> > I'd like to thank the authors for adding three baselines in Table 1 and preparing the responses to my concerns on GDL.
> >
> > However, I am not very satisfied with the answers for Q1 and Q3. Overall, I still believe this work is near the borderline considering the technical novelty and meaningness to the BCI community. I will keep my score (4) unchanged.

---

> > > ### Author Response · Authors · 2022-08-09
> > > **Thanks to Reviewer Ad4a for the responses**
> > >
> > > We thank for the reviewer's feedback regarding our revision. It is unfortunately that our answers for Q1 and Q3 were not satisfying enough. In our work, the properties of manifold such as the affine invariance of Riemannian manifold can handle the non-stationarity EEG properly. Therefore, we assume that the use of GDL is an inevitable step toward high-performance EEG decoding.

---

### Official Review · Reviewer_xJ3M · 2022-07-19

**Rating:** 7
**Confidence:** 5
**Soundness:** 3 good
**Presentation:** 3 good
**Contribution:** 3 good

**Summary:**

This study presented a novel method for the brain-computer interface, where a manifold attention mechanism is applied to Riemannian SPD. The results showed significant improvement over the SOTA methods on two typical datasets (motor imagery and SSVEP). The paper is clearly written with adequate analysis. The explanations provide interesting insights.

**Questions:**

Q1: The attention scores are highest for the first part of the MI data and moderate for the middle part, are they linked to the ERD/ERS?
Q2: For SSVEP, why the attention scores should be different for different parts of the data if the underline signals are the "steady" visual responses from the EEG?

**Limitations:**

The proposed method provides a novel treatment to BCI with promising performance. More experiments on different datasets should be tested to confirm its generalizability.

**Strengths And Weaknesses:**

Strengths:
- This study is a new development of geometric deep learning for the brain-computer interface.
- The results are promising.

Weakness:
- The interpretations should provide more in-depth analysis to provide deep insight. For example, it shows the attention scores are high for the first part of the MI data and moderate for the middle part. Does it link to ERD/ERS? For SSVEP, why the attention scores should be different for different parts of the data if the underline signals are the "steady" visual responses from the EEG?

---

> ### Author Response · Authors · 2022-08-02
> **Response to Reviewer xJ3M**
>
> We appreciate your comments and suggestions has largely refurbished our paper.
> Thanks for the insightful comment. We believe that the neuroscientific explanation of the attention score would be an interesting issue for us to explore. We decided to list this task as the future work in appendix A.7 since we believe the neuroscientific knowledge should be check via the biological experiment and special design, which is already beyond out of the scope of our work. We hope this issue can be well clarified in the neuroscience in the future. But for the Euclidean-based self-attention work, we recommend [1] for reviewer to refer to. In our opinion, although the dataset that this paper adopted is sleep dataset, there are some interesting visualization shown in their work.
>
> [1] Sleeptransformer: Automatic sleep staging with interpretability and uncertainty quantification by Huy Phan et al.
>
> 1) Attention score: The attention score refers to the similarity between epochs within an EEG trial data. As we observe a consistency cross EEG datasets that higher attention scores lie in earlier epochs, this may infers that the attention module relies largely on the similarity to the early stage of a trial, which is analogous to baseline correction, a major common procedure in conventional EEG signal processing. Please refer to Section 4.3.
> 2) Attention score and SSVEP: Thank you for the opportunity for us to clarify. According to the computation of attention score, we believe that the attention scores present how similarity across time affect the recognition of the data. Therefore, the attention scores reveal the underlying data processing within the attention module. On the other hand, the "steady" of the SSVEP describes the steady pattern evoked by repeated visual stimuli.

---

### Author Response · Authors · 2022-08-02
**Summary of response to reviewer comments**

The authors would like to thank all reviewers for their constructive comments that have significantly helped improving this paper. Some of the reviewers pointed out major strengths of this work, i.e. our proposed GDL-based model, mAtt, can enhance the accuracy of EEG decoding, and, unlike most of the counterparts, its performance can be generalized to various types of EEG data. Based on the promising evaluation results, this novel technique can shed light on the milestone of GDL-based EEG decoding. We have responded to all of the comments carefully and made all of the necessary modifications within a reasonable time frame. The revised manuscript contains modifications for explicit explanations as well as a new version of convincing experimental results. We have expanded the selection of baseline leading models to validate the performance of our proposed method. We also incorporated an additional benchmark time-synchronous EEG dataset to assess the generalizability of the model performance. Moreover, additional ablation analysis has been conducted to dismantle the component within our mAtt model to assess the effect of utilizing manifold-based attention in the model. In sum, our proposed model, mAtt, has achieved the leading performances in both motor-imagery and SSVEP EEG decoding and a satisfying performance in ERN EEG detection.

---

### Meta-Review · Area_Chair_4P3C · 2022-08-24

**Recommendation:** Accept
**Confidence:** Certain

**Metareview:**

The paper proposes a manifold attention network (mAtt), that leverages the properties of the SPD manifold in order to perform spatiotemporal analysis on EEG signals. The results showed significant improvement over the SOTA methods on two typical datasets (motor imagery and SSVEP). The paper is clearly written with adequate analysis. The idea of representing EEG signals as covariance matrices and then projecting them on the SPD manifold has been widely used. The novelty of the approach lies mainly in the application of the attention mechanism on the SPD manifold. The authors' responses have successfully addressed some reviewers' concerns and provided additional (supportive) experimental results, which convinced reviewers to update their evaluations. Some minor concerns about why geometric deep learning is necessary in EEG classification and the clarity of the paper could be further improved. The authors are encouraged to take reviewers' detailed comments into account in the final version.

**Award:**

No

---

### Decision · Program_Chairs · 2022-09-14

Accept